# The Genome-Wide Characterization of Alternative Splicing and RNA Editing in the Development of *Coprinopsis cinerea*

**DOI:** 10.3390/jof9090915

**Published:** 2023-09-09

**Authors:** Yichun Xie, Po-Lam Chan, Hoi-Shan Kwan, Jinhui Chang

**Affiliations:** 1State Key Laboratory of Agrobiotechnology, Food Research Center, School of Life Sciences, The Chinese University of Hong Kong, Shatin, New Territories, Hong Kong SAR, China; xieyichun50@link.cuhk.edu.hk; 2Food Research Center, School of Life Sciences, The Chinese University of Hong Kong, Shatin, New Territories, Hong Kong SAR, China; 3Department of Food Science and Nutrition, and Research Institute for Future Food, The Hong Kong Polytechnic University, Hong Kong SAR, China

**Keywords:** *Coprinopsis cinerea*, life history, RNA editing, alternative splicing, transcriptomic divergence, adaptation

## Abstract

*Coprinopsis cinerea* is one of the model species used in fungal developmental studies. This mushroom-forming *Basidiomycetes* fungus has several developmental destinies in response to changing environments, with dynamic developmental regulations of the organism. Although the gene expression in *C. cinerea* development has already been profiled broadly, previous studies have only focused on a specific stage or process of fungal development. A comprehensive perspective across different developmental paths is lacking, and a global view on the dynamic transcriptional regulations in the life cycle and the developmental paths is far from complete. In addition, knowledge on co- and post-transcriptional modifications in this fungus remains rare. In this study, we investigated the transcriptional changes and modifications in *C. cinerea* during the processes of spore germination, vegetative growth, oidiation, sclerotia formation, and fruiting body formation by inducing different developmental paths of the organism and profiling the transcriptomes using the high-throughput sequencing method. Transition in the identity and abundance of expressed genes drive the physiological and morphological alterations of the organism, including metabolism and multicellularity construction. Moreover, stage- and tissue-specific alternative splicing and RNA editing took place and functioned in *C. cinerea*. These modifications were negatively correlated to the conservation features of genes and could provide extra plasticity to the transcriptome during fungal development. We suggest that *C. cinerea* applies different molecular strategies in its developmental regulation, including shifts in expressed gene sets, diversifications of genetic information, and reversible diversifications of RNA molecules. Such features would increase the fungal adaptability in the rapidly changing environment, especially in the transition of developmental programs and the maintenance and balance of genetic and transcriptomic divergence. The multi-layer regulatory network of gene expression serves as the molecular basis of the functioning of developmental regulation.

## 1. Introduction

The fungal kingdom is highly diverse and plays essential roles in the ecosystem. *Basidiomycota* is the second-largest phylum and the most evolutionarily advanced group of fungi [1,2]. The basidiomycetous species have complicated life histories that can be divided into several stages [3]. Fruiting bodies of different species in *Basidiomycota* show high diversity, and their formation is generally the most complex developmental process in the fungal life cycle [3,4,5]. For *Agaricales*, a group of typical mushrooms, different types of monocellular or multicellular structures are formed in different developmental paths [6,7]. Apart from fruiting body formation, these species have more developmental destinies, including asexual spore formation, persistent resting structure formation, and pseudorhiza formation [3]. The developmental process of fungi is regulated via both cytological events and environmental factors.

*Coprinopsis cinerea* is a commonly used model species in fungal developmental studies. It is a typical mushroom-forming fungus with several alternative developmental destinies [3,7]. The growth and development of *C. cinerea* are regulated via both internal physiological factors and external environmental factors, including light, temperature, and nutrients [8]. Under different environmental conditions, the fungus chooses one of the developmental strategies from vegetative growth, asexual reproduction, sexual reproduction, or stress-resistant structure formation (Figure 1). Vegetative mycelia expand on the nutrient-rich substrate at 37 °C with continuous darkness; asexual reproduction of oidiation is induced under the nutrient-rich environment at 37 °C with illumination; sexual reproduction happens when the fungus is under nutrient depletion and exposed to low temperature with light–dark cycles; the stress-resistant structure formation of sclerotia develop under nutrient depletion and continuous darkness [3,7]. In the homokaryotic strain *C. cinerea A43mutB43mut pab1-1* #326, fruiting bodies develop without mating due to mutations in the mating-type genes [9,10]. These features eliminate the impact of allele diversity and provide the chance to investigate dynamic regulations in *C. cinerea* development in a clear genetic background [11]. The *C. cinerea* #326 strain displays similar features to the unmutated strains during development, including spore germination, hyphal branching, vegetative growth, fruiting, oidiation, and sclerotia formation [10,12,13].

Gene expression is carefully regulated during fungal development. Various mechanisms and molecules participate in the pre-, co-, and post-transcriptional regulatory processes [14,15]. Transcription factors are key regulators of gene expression. In *C. cinerea*, the mating-type A locus genes encode homeodomain transcription factors. They are responsible for mating recognition and processing and promote the expression of down-stream targets to conduct nuclear migration and clamp formation [16]. During the development of the fruiting body, the GATA transcription factor CcnsdD2 regulates the hyphal knot initiation under light or dark conditions, and guides the formation of the fruiting body, dark stipe, or sclerotium [17]. In addition to transcription factor-binding, DNA methylation also affects the transcription. In *C. cinerea*, methylation suppresses the expression of paralogous multicopy genes in oidia in a manner of epigenetic marks [18]. Moreover, RNA interference, which alters the gene expression via small RNA and unpaired DNA, has been found in several fungi [19,20]. MicroRNA-like small RNAs (milRNAs) in *C. cinerea* regulate the developmental transitions during the formation of the fruiting body and the germination of basidiospores [21,22]. While the above processes mainly affect the gene expression levels, alternative splicing (AS) could diversify both the sequence and abundance of transcripts [23,24]. Although the developmentally regulated transcript isoforms are commonly found in fungi [25,26], how they benefit the organism in development and adaptation requires further investigation.

RNA editing (RE) is an RNA modification process that takes place at the co-transcriptional or post-transcriptional level. Without disturbing the genomic hardware, changes at the RNA level provide phenotypic flexibilities. It has been found that RNA editing displays a trade-off between transcriptome plasticity and genome evolution [27,28]. Biochemical modifications on nitrogenous bases of the nucleotide can lead to base substitutions in RNA. For example, in the two major RNA editing types, adenosine or cytidine deaminases catalyze the specific base deamination, causing A-to-I or C-to-U editing [29]. As the nucleotide sequence is changed, this form of editing would affect amino acid sequences, molecular recognition and interactions, and gene expression regulations, causing rapid changes in the transcriptome and proteome [30]. RNA editing has been widely reported in animals and plants [31,32]. In the fungal kingdom, such events were first identified in the fruiting bodies of *Ganoderma lucidum* [33]. In *Ascomycota*, RNA editing occurs in a stage-specific manner during sexual reproduction in the genera *Neurospora* and *Fusarium* [34,35], bringing adaptive advantages to the organism with both edited and unedited versions of transcripts and protein products [36]. Specifically, RNA editing on the PUK1 transcript was proven to be essential for the sexual reproduction in *Fusarium graminearum* [35]. RNA editing has been found to be abundant and widespread in the above-mentioned species and regarded as a novel layer in the gene expression regulatory network in *Ascomycetes* development [37]. However, the occurrence, characteristics, and functional effects of RNA editing in *Basidiomycetes* remain unclear [38,39].

Although several studies have profiled the transcriptional regulations in mushroom formation of *C. cinerea* [25,38,40], knowledge on other developmental paths is still rare. Our previous study showed that carbohydrate metabolic flux and gene expression profiles diverged among the developmental paths and processes [7,22]. We further questioned the effect of RNA modifications on *C. cinerea* development. Additionally, whether different regulatory strategies drive the specific developmental process is largely unknown. In this study, we revisited the transcriptome profiles in different developmental processes and stages of *C. cinerea* #326. We aimed to (i) characterize the dynamics of alternative splicing and RNA editing during the development of *C. cinerea*; (ii) uncover the functional impacts of these transcriptome modification events; and (iii) clarify the adaptive features of modifications and regulations in the transcriptome from an evolutionary perspective. Our results shall provide new insights into the reconstruction of a multi-layer gene expression regulatory network in the development of complex multicellularity in fungi.

## 2. Materials and Methods

### 2.1. Fungal Cultivation, Sample Collection, and High-Throughput Sequencing

The homokaryotic strain *C. cinerea A43mutB43mut pab1-1* #326 was chosen for this study. The working stock of the fungus was obtained by growing the culture on solid YMG medium (composed of 0.4% yeast extract, 1% malt extract, 0.4% glucose, and 1.5% agar, and 36 g medium per 90 mm diameter petri dish) at 37 °C with continuous darkness for 5 days, as previously described [7]. Details of the strain cultivation and sample collection for each developmental path and stage are shown in Table 1.

The transcriptome data of basidiospores (BS), half-germinating basidiospores (BS12h), fully germinated basidiospores (BS24h), and vegetative mycelia (Myc), oidia-forming mycelia (Oidia), sclerotia-forming mycelia (Scl), and mycelia with hyphal knots (Knot) were described in our previous developmental studies [7,22]. In addition, data of two other stages of fruiting body formation were collected, including the primordia undergoing meiosis (Pri) and young fruiting bodies undergoing spore formation (YFB). They were incubated the same as the hyphal knots, except for the longer exposure period under 12 h:12 h light–dark cycles at 28 °C. The sampling time was determined through dissection and microscopic examination (Appendix A). Mycelia with hyphal knots were harvested after one light–dark cycle, while primordia and YFBs were harvested after 6 cycles and 6.5 cycles, respectively. Total RNA from the caps of primordia and YFBs was isolated using the RNeasy Plant Mini Kit (Qiagen, Germany), following the manufacturer’s instruction after microscopic examination. Approximately 5 µg of the total RNA of each sample was sent to the Beijing Genomics Institute (BGI, Shenzhen, China) for library construction and sequencing. RNA libraries were prepared using the TruSeq RNA Sample Prep Kit v2 (Illumina, San Diego, CA, USA) and sequenced with Illumina HiSeq^®^ 4000 at the 2 × 150 bp paired-end read mode, the same as was conducted in our previous studies [7,22]. Three biological replicates were collected for each stage in RNA-seq. Illumina short-read DNA sequences were acquired from our previous work of de novo genome assembly on strain #326 [11] to serve as the supporting genomic sequence information for the determination of RNA editing. In this study, all DNA and RNA samples were collected from the same batch of culture. Sequencing quality control is summarized in Appendix A. To better perform developmental comparisons, we further grouped the nine stages into the five following processes: (i) spore germination: BS-BS12h-BS24h; (ii) oidiation: Myc-Oidia; (iii) sclerotia formation: Myc-Scl; (iv) fruiting: Myc-Knot-Pri; and (v) sporulation: Pri-YFB-BS.

### 2.2. Read Mapping and Counting

Sequencing reads were quality controlled and filtered using fastp with default settings of a sliding window size of 4 nt and an average base quality of 20 [41]. DNA and RNA reads were aligned to the reference genome of #326 (GenBank: GCA_016772295.1) [11] using Bowtie2 (version 2.2.6) [42] and Hisat2 (version 2.1.0) [43], respectively. Duplicated reads were marked and removed using the “MarkDuplicates” function in Picard tools (version 2.9.0, http://broadinstitute.github.io/picard, accessed on 9 August 2023). DNA and RNA libraries were then proceeded to the in-depth analyses.

Gene expression levels were calculated using Stringtie (version 2.1.7) [44] with the reference gene annotation of *C. cinerea* #326 [11]. The count matrix was then fed to the R package ‘edgeR’ (version 3.36.0) [45] for expression analysis and presented using the trimmed mean of M-values (TMM). Genes with a count-per-million (CPM) over one in at least two out of three replicates per stage were regarded as expressed genes. For each gene, *i*, the relative expression level (*RE*) of each sample, *k,* was calculated as $RE_{\left[i,k\right]}=\frac{Gene_{\left[i,k\right]}-Gene_{\left[i\right]min}}{Gene_{\left[i\right]max}-Gene_{\left[i\right]min}}$ using log2-transformed TMM values. Differentially expressed genes (DEGs) were determined with the threshold of |log2 (Fold change)| > 2, which was a four-fold change, and the adjusted *p*-value threshold of lower than 0.05.

### 2.3. Identification of Alternative Splicing

Alternative splicing (AS) events were identified using CASH (version 2.2.0), a self-construct alternative splicing detector [46]. The alignment file in bam format and gene model annotation file in gff3 format were supplied into CASH, and biological replicates were considered. At least 10 reads in the specific stage supported the junction, and only events with a total coverage of over 25 reads were considered. Sequences found in intergenic regions or overlapping gene regions were removed. To eliminate the occurrence of false positives caused by alignment error, AS sites with mismatches, insertions, or deletions near an overhang of a read alignment junction within 6 bases were also removed [47,48]. Percent spliced in (PSI) was calculated for each of the nine stages. Rare isoforms reached the expression of 5 and an abundance ratio of over 5% in at least one stage. Differential AS events were determined with the cut-off of |delta PSI| > 0.1, *p*-value < 0.05, and false discovery rate < 0.05 using the Benjamini–Hochberg method.

### 2.4. Identification of RNA Editing

RNA editing candidates were called using the python script REDItoolBlatCorrection.py and REDItoolDnaRna.py in REDItools (version 1.0.4) [49] with the following thresholds: (i) to minimize false discovery caused by sequencing and mapping errors, the minimum read quality and read mapping quality are set to 25 in both DNA and RNA, and multi-hit reads are excluded; (ii) to ensure the significance of occurrence of editing events, at least 3 reads should support the variant, with a minimum frequency of 3%, and a minimum total coverage of 10 reads; (iii) excluding positions not supported by or with changes in DNA-seq to avoid false-positive events caused by genomic variants; (iv) excluding positions near the splice site within 4 bases to avoid mismatch caused by the misalignment of RNA; and (v) editing events present in at least 2 biological-replicated samples of any specific stages.

### 2.5. Reverse Transcription and PCR Amplification

To further validate the gene expression levels, AS, and RE events, cDNA was synthesized from the total RNA with an anchored-oligo(dT)18 primer and a random hexamer primer using the Transcriptor First Strand cDNA Synthesis Kit (Roche, Germany). The template–primer mix was denatured at 65 °C, and the reverse transcription reaction was incubated as follows: 10 min at 25 °C, 30 min at 55 °C, and 5 min at 85 °C. One μg of RNA was inputted into each 20 μL reaction.

PCR amplification was performed using cDNA or gDNA as templates with the KAPA HiFi HotStart ReadyMix PCR kit (Roche, Germany) and the following program: 95 °C for 3 min, followed by 30 cycles of 98 °C for 20 s, 65 °C for 20 s, 72 °C for 45 s, and 72 °C for 2 min. PCR products were detected on a 1.5% agarose gel and purified with the MEGA quick-spin Plus Fragment DNA Purification Kit (iNtRON Biotechnology, Seongnam-si, Republic of Korea). Sanger sequencing on the PCR products of DNA and cDNA was performed by Beijing Genomics Institute (BGI, Shenzhen, China).

Quantitative real-time PCR (qRT-PCR) was used to quantify the transcripts. Tests were carried out using the SsoAdvanced^TM^ Universal SYBR^®^ Green Supermix (Bio-Rad, Hercules, CA, USA) with the Applied Biosystems^TM^ 7500 fast Real-Time PCR System (Applied Biosystems, Waltham, MA, USA). PCR reactions were performed as follows: 30 s at 95 °C, followed by 40 cycles of 15 s at 95 °C and 30 s at 60 °C, and the instrument default setting on melt curve analysis. The internal control used for this procedure was 18S rRNA. The primers used in this study are listed in Appendix A.

### 2.6. Functional Annotations and Analyses

Functional annotations of genes were based on the reference genome of #326 (GenBank: GCA_016772295.1) [11]. Functional enrichment analyses were carried out using the ‘compareCluster()’ function in the R package ‘clusterProfiler’ (version 4.2.2) [50], with ‘pvalueCutoff = 0.2, pAdjustMethod = “BH”, qvalueCutoff = 0.2’. The coding potential of RNA isoforms was predicted by retrieving the exon sequences and translating them into protein sequences. SnpEff version 5.1 [51] was used to predict the functional impact of RNA editing. Interactions between the milRNA and the transcripts were predicted with RNAhybrid (version 2.1.2) [52]. Phylotranscriptomic analyses were performed as previously described [53,54]. For each category *i* in TAI, TDI, AS, and RE, the relative value (*RV*) of stage *k* was normalized between 0 and 1, and was calculated as $RV_{\left[i,k\right]}=\frac{Value_{\left[i,k\right]}-Value_{\left[i\right]min}}{Value_{\left[i\right]max}-Value_{\left[i\right]min}}$. Heatmaps were generated using the ‘pheatmap()’ function in the R package ‘pheatmap’ (version 1.0.12) [55]. All statistical analyses were conducted under the R (version 4.1.2) environment [56].

## 3. Results and Discussion

### 3.1. Overview of C. cinerea Developmental Transitions

To investigate the physiological and transcriptional changes in the life cycle of *C. cinerea*, we cultivated and induced the strain into vegetative growth, sexual reproduction, asexual reproduction, and persistent resting by altering the environmental conditions of temperature and light period (Figure 1). The amount of nutrients in the culture media decreased along with the growth processes. When the mycelia fully covered the agar surface (90 mm diameter petri dish) at 5.5 days post-inoculation (dpi), less than 40% of the reducing sugar was available in the sediment for vegetative growth under continuous darkness or oidiation under light at 37 °C [7]. The fungus culture then entered the fruiting process or sclerotia formation under specific combinations of temperature and lightening. After 14 days of incubation under continuous darkness at 37 °C, sclerotia were formed and fully matured into brown or dark brown round shapes of ~1 mm diameter. Meanwhile, nutrient-depleted cultures that were shifted to 12 h:12 h light–dark cycles at 28 °C underwent the fruiting body formation process. Fruiting body initials were found on the culture surface within 3 light–dark cycles, and the stipe and cap were able to be distinguished in primordia in 4–5 light–dark cycles. The earliest and latest maturation of fruiting bodies can show a time difference from one to two days. Based on this case, we not only determined the developmental stages through the external features, but also the cell and tissue’s structure using microscopic examination. In the cap of primordia, bud-shaped basidia were well-ordered on the hymenia (Appendix A). These samples were regarded as the primordia undergoing meiosis. After one more light period of 12 h, the sterigmata were visible and the spores were enlarged (Appendix A). It took these spores another 12 h in darkness to fully mature and be discharged from the cap. When the spores were moved to fresh media, which were rich in nutrients with hydration, spore germination was activated. The volume of the spores soon enlarged due to water uptake and cell wall growth. At 12 h after spore transfer, germination tubes that were longer than the half-size of the spores could be observed in half of the culture. After incubation for 24 h, septa occurred in the germination tubes, and hyphae branched and connected with each other [22]. A hyphal network was formed in the mycelia in the culture hereafter.

### 3.2. Gene Expression Profiles in Different Developmental Trajectories

We obtained the transcriptomes of nine stages, namely basidiospores (BS), half-germinating spores (BS12h), fully germinated spores (BS24h), vegetative mycelia (Myc), oidia-forming mycelia (Oidia), sclerotia-forming mycelia (Scl), mycelia with hyphal knots (Knot), primordia undergoing meiosis (Pri), and young fruiting bodies undergoing spore formation (YFB), with three biological replicates. The gene expression profile was filtered through the expression value CPM over one in at least two replicates. A total of 12,450 genes were detected during *C. cinerea* development, and the number of detected genes varied from 10,043 (BS) to 11,396 (Knot) (Appendix A). All gene expression levels are shown in TMM values in Appendix A. Gene expression levels of these stages were compared against one another. The proportion of DEGs varied from 1.7% (Myc vs. Oidia) to 41.7% (YFB vs. BS) (Appendix A). Most DEGs displayed a log2 fold change between −8 and 8 (Appendix A).

To have a global view on gene expression regulation, DEGs were extracted and clustered based on their relative expression levels (Figure 2). Genes of the six different clusters were enriched to specific functional terms by means of KOG (Appendix A), KEGG (Appendix A), and GO (Appendix A and Appendix A) annotations. In gene cluster 2, genes were active in mycelia-type samples, which included vegetative mycelia, oidia-forming mycelia, sclerotia-forming mycelia, and mycelia with hyphal knots. These genes functioned in basic metabolism and maintenance of cellular structure. Genes in cluster 4 were also active in mycelia-type samples; however, they were developmentally regulated during fruiting and sporulation, with primary functions in cell cycle control, cell division, and chromosome partition, as well as the biogenesis of fungal cell structures and morphogenesis. Genes of cluster 1 showed higher expression levels in primordia and young fruiting bodies, and carbohydrate metabolism and MAPK pathways were also involved. On the contrary, the genes in cluster 5 were more active in spore maturation and fresh spores, but silent in other processes. These genes were involved in RNA transcription and protein translation. In cluster 3, genes were annotated with varied metabolic roles that were related to small molecule metabolisms, and they contributed to the early germination responses. Finally, genes in cluster 6 were intensively regulated during spore germination, and their expressions were low in mycelia and fruiting bodies. Such genes are involved in the polarized hyphal growth and formation of mycelia, but not in advanced multicellular structures, like mushrooms.

### 3.3. Alternative Splicing Landscape in C. cinerea

Alternative splicing events were identified in all stages (Appendix A). Restricted by at least 10 supporting reads and a minor isoform abundance ratio of over 5%, a total of 23,113 AS events were screened out from nine developmental stages (Table 2). These events were located in 3411 genes with different regulatory features (Appendix A and Appendix A). The number of AS events was similar in most stages, except during the basidiospore germination process when basidiospores carried the highest number of AS events, and soon decreased within 24 h. Intron retention and alternative 5′/3′ splice events were the major AS types, and both took up over 40% of the AS events across all stages (Appendix A). The distribution of percent spliced in (PSI) differed from stage to stage. Primordia, young fruiting bodies, and basidiospores showed significantly higher PSI values than others by 0.1 (Appendix A and Appendix A). Further, PSI values also varied by splicing types along the life history (Appendix A). When looking into the eight splicing types, the distribution of PSI values was not significantly different across the stages, except for the intron retentions (Appendix A).

AS events frequently occurred during *C. cinerea* development, but a large proportion of AS regions obtained similar PSI values in different stages. Using the criteria of |delta PSI| over 0.1, it was found that 2723 out of 6839 AS regions (39.8%) were differentially spliced among stages (Figure 3a). We then linked the gene expression level and PSI value together to further explore the global features of AS on gene regulation. No correlation was identified between the AS levels and expression levels (Appendix A), but genes that were without significant expression changes tended to have stronger splicing alterations (Appendix A). When performing one-to-one comparisons among the stages, the range of delta PSI and log2 fold changes varied (Appendix A). Stages with greater differences in the complexity of multicellularity displayed a broader area of expression and splicing differences, which indicated a dramatic turn-over of the transcriptome during fungal development via gene identity and composition.

### 3.4. Alternative Splicing Generated Stage-Specific Isoforms during Development

AS profiles varied by stages along the *C. cinerea* life history (Appendix A). The abundance of isoforms in the basidiospore was most different from the other eight stages, with a range from 1151 to 1413 AS regions differentially spliced. Fewer differential splicing events were detected between stages of the same continuous development. In addition, similar to the gene expression profile, AS profiles of mycelia-type samples were also close to one another (Appendix A). Only 230 AS regions had a PSI change of over 0.1 between the vegetative mycelia and oidia (Appendix A).

By grouping the nine stages into five developmental processes, 1030 AS events were only found to be differentially regulated in a specific process (Figure 3b). Intron retention and alternative 5′/3′ splice events remained the major splicing types (Figure 3c). During fruiting and sporulation, 249 and 513 AS regions were specifically regulated, with KOG terms enriched in “RNA modification and processing” and “transcription”, respectively (Appendix A). In sclerotia, 45 regions were specifically regulated, typically involved in protein metabolism and secondary metabolites (Figure 3d). Additionally, AS genes that were specific to spore germination were mainly cellular processing and signaling, specifically, “intracellular trafficking, secretion and vesicular transport” (Appendix A).

### 3.5. Broad Effects of Developmentally Regulated Alternative Splicing in C. cinerea

During the development of the fungus, several genes were differentially expressed and spliced during fruiting body formation and sporulation. (1) Spliceosomal U1 snRNP C (CC2G_010208): intron retention between the third and fourth exon would include 18 other amino acids, and AS sites between the fifth and sixth exon would cause early termination of the protein (Figure 3e). These events could change the protein folding and molecule-binding activity of the spliceosome and the splicing activities of downstream targets. (2) Rho GTPase-activating protein (RhoGAP, CC2G_009456) varied by 19 serine-rich amino acid sequences before and after intron retention, creating a low complexity region to the protein (Figure 3f). This structure variant might affect the protein–protein interaction, such as with the Rho GTPase modification and vacuolar protein sorting process. (3) STE/STE11 protein kinase: the intron retention between the fourth and fifth exon did not change the catalytic domain but affected the helix-turn-helix folding. The abundance of two isoforms was similar in vegetative mycelia and hyphal knots, and the exclusion form of the transcript was dominant in sclerotia, while the inclusion form was dominant in oidia, primordia, and young fruiting bodies (Figure 3g). (4) In silico analysis predicted that AS on the cassette exon of regulatory proteins, protein kinase PKL/CAK/ChoK (CC2G_001453), would cause the early termination of translation, and lead to failure on the assembly of protein subunits. (5) A cassette exon inclusion form of the RNA sequence of CAMK/CAMK1 protein kinase (CC2G_010331) dominated the transcript during sporulation. Although AS adds protein sequences to the catalytic domain of the protein kinase, which is recognized as a member of the PKc-like superfamily, and modifies its alpha helix region, this AS event was predicted to not affect the identification of the conserve domain.

AS is found in a variety of fungal species and is important to the establishment of multicellular complexity [25,57]. This posttranscriptional modification would primarily generate transcript isoforms [58]. Depending on the AS location, it rewrites the protein coding sequence, or regulates the expression level via changing the dynamics of noncoding RNA binding, protein binding, or the nonsense-mediated RNA decay pathway [59,60]. Previous studies have emphasized the role of AS in the multicellularity formation of fruiting bodies [25,61], whereas our results showed that AS not only occurred during fruiting, but also during sclerotia formation and spore germination. These findings revealed the general effect of AS in increasing the complexity of structures in fungi [62]. In fresh basidiospores, RNA was synthesized before and soon after the basidiospores dispatching from the mature fruiting bodies [22,63]. Our results showed that basidiospores had the highest number of AS sites among all developmental stages, reflecting the diversification of the transcriptome and functional proteome. In *Aspergillus nidulans*, it was shown that the differences in transcriptomes affected the survival of conidia under different environments and the phenotype during and after conidia germination [64]. We suggested that the widespread AS in basidiospores would provide ready-to-use materials in survival and morphogenesis; that is, playing important roles in the early responses against the changing environments, and in the development from single-cell to multicellular structures. Additionally, our results uncovered that a number of differentially spliced genes were regulators of signal transduction and RNA/protein modification and processing. These different isoforms would have different subcellular localizations and different enzymatic efficiency, further diversify the downstream targets [65,66]. AS was revealed to have a broad and flexible mediation potential in the decision and regulation during fungal development [58,67,68].

### 3.6. RNA Editing Happened in a Stage-Specific Manner along the Development of C. cinerea

Our screening of the transcriptomes identified 217 RNA editing sites in *C. cinerea* development (Table 2 and Appendix A). Sclerotia had the most RNA editing events among all stages, in terms of both editing sites and the number of events per million mapped reads (Figure 4a). U-to-C editing dominated the editing types found in *C. cinerea*. Three other editing types, G-to-A, C-to-U, and A-to-I(G), took up 5–9% of editing events and sites (Appendix A). Eight other editing types were less commonly detected, and A-to-U editing was only found in three positions. Most of these RE events showed editing levels of less than 0.15, and the average editing level of all events was 0.081 (Appendix A). The editing levels were not much different across stages, with basidiospores and oidia having slightly higher mean editing levels of 0.143 and 0.105, respectively (Figure 4b and Appendix A). The distribution of editing levels of twelve editing types was found to be significantly different (*p* < 0.001, Figure 4c, Appendix A).

A large proportion of RE sites was not shared among the tissues and stages. Among 217 RE sites, only 163 occurred in one specific stage (Appendix A). The editing profiles displayed a relationship between the RE and the robust development that corresponded to the nutrient and morphogenesis, indicating the stage-specific pattern and dynamic RE in *C. cinerea* development (Figure 4d). Sclerotia formed as stress-resistant multicellular structures under a stressful environment with nutrients depleted. Sclerotia had 97 stage-specific RE sites (Appendix A), shaping the distinct editing landscape and separating themselves from other samples and clades. The vegetative mycelia, oidia-forming mycelia, and mycelia with hyphal knots were clustered together, with about one-third to half of the RE sites shared among these stages (Appendix A). These stages all needed to adapt to and grow under various environments. Despite primordia and young fruiting bodies undergoing intensive morphogenesis, RE was not common (Appendix A). The serially sampled transcriptome profiles of fruiting body development in *C. cinerea* [40] displayed similar RE features (Appendix A, Appendix A). Stages within shorter sampling intervals shared more common RE sites, but the most events were stage- and tissue-specific (Appendix A). During spore germination, intensive morphogenesis and structure development changed single-cell spores into mycelia. They had the lowest RE intensity and the least number of shared RE sites (Appendix A), indicating that the conserved regulations on gene expression instead of the robust RE events determined the developmental process. Moreover, the RE profiles of basidiospores were closer to the fruiting bodies than to the mycelia.

### 3.7. RNA Editing Diversifies the Regulation of Gene Expression

To clarify the genetic locations and functional impacts of RE, these RE sites were further mapped against the gene model annotations (Appendix A). A total of 89 editing sites (41.0%) were found in intergenic regions, while 128 other editing sites were located at the coding regions (CDS) or untranslated regions (UTR) of 114 genes, and none was found in the intron (Figure 5a). The genetic locations of RE were not significantly different among the nine stages (Figure 5b). Editing levels were different among the four genetic locations, and those that happened on CDS showed lower editing levels, with a few extreme cases (Figure 5c). For the 58 editing sites that were found in CDS, more than half of them sat on the third letter of the codon (Figure 5d). We found 29 out of 58 sites to be synonymous, with no differences in the protein sequence. On the other hand, 29 RE sites would change the amino acid sequence, create the premature termination of the protein, or cause the loss of the stop codons (Figure 5e). For example, scaffold_1:2116854C>G editing was only detected in sclerotia. The change from UAC to UAG would turn the 577th tyrosine into a stop codon, losing 137 amino acids, breaking up the WD40 repeat sequence structure of the Trp-Asp repeat-containing protein. The missense sclerotia-specific variant of ubiquitin and ribosomal protein S27a (CC2G_012131, scaffold_6:2283059U>C) changed the conserved functional domains. A developmentally regulated U-to-C editing (scaffold_11:1716350, CC2G_003350) site was also identified. Editing was detected in vegetative mycelia, oidia, sclerotia, and hyphal knots, but not in the other five stages, with the editing levels varying from 0.03 to 0.7 (Figure 5f and Appendix A). This editing caused synonymous changes on this leucine-rich hypothetical protein. Apart from editing in CDS, 70 RE sites were located at the UTR of 66 genes. In silico analysis was performed on the interactions between the known milRNA [22] and these UTRs. The binding of two RNA molecules was altered after the base substitutions (Figure 5g). Therefore, RE can also cause a gain or loss in milRNA–UTR interactions and affect the performance of transcripts.

Functional enrichment analysis indicated that the edited genes corresponded to posttranslational modification, protein turnover, and chaperon- and ubiquitin-mediated proteolysis (Appendix A). Editing events were also found in several protein kinases and nuclear proteins (Appendix A), suggesting that RE might have functional effects on the core regulatory factors of gene expression. The widespread RE in the fruiting bodies of *G. lucidum* was enriched in transcriptional regulation, wood degradation, and secondary metabolite biosynthesis [33]. However, the *G. lucidum* study only included the fruiting body; the developmental effects were not evaluated. Our genome-wide profiling on nine developmental stages characterized RE along the life cycle of *C. cinerea* and is one of the pioneer studies on developmentally regulated RE in *Basidiomycetes*. Unlike the *Ascomycetes*, a data revisiting study on the *C. cinerea* transcriptome atlas concluded that *C. cinerea* had no evidence of RE during sexual development [39]. Here, our analysis on transcriptomes of several developmental paths and a previous fruiting body developmental study [40] both showed that *C. cinerea* undergoes RE during its development. Sclerotia, which were induced under a stressful environment, displayed the greatest extent of RE enriched in the genes of RNA processing and protein binding, which are essential for metabolic regulation. In addition, both sexual spores (basidiospores) and asexual spores (oidia), which are expected to respond to a broad range of environmental conditions, showed higher editing levels than other stages and were enriched in signaling genes. Moreover, RE was highly stage-specific in *C. cinerea*. Together, these results suggested that RE can provide the molecular basis for a diversified regulation during asexual development and facilitate the better adaptation to changing environments by fungi [36,69,70].

### 3.8. Putative Catalytic Genes of RNA Editing in C. cinerea

RNA editing has been reported in both *Ascomycetes* and *Basidiomycetes* [39,71]. In several Ascomycetes fungi, such as *Fusarium graminearum* and *Neurospora crassa*, editing occurred at specific stages during sexual reproduction, with strong A-to-I(G) editing preferences, and such modifications were essential for perithecia development [34,35]. However, our results, and previous studies on *Ganoderma lucidum*, found that the A-to-I(G) editing was not preferred in *Basidiomycetes*, but together with C-to-U, G-to-A, and U-to-C, as the four major types of editing. Although the homologs of ADATs have been identified in a variety of species, such as *Saccharomyces cerevisiae*, *Schizosaccharomyces pombe*, *F. graminearum*, *Aspergillus nidulans*, and *Neurospora crassa* [35], the mechanism of RNA editing in fungi remains unclear [72], especially regarding non-A-to-I(G) editing.

To explore the molecular mechanisms of RNA editing in *C. cinerea*, genes with RNA editing catalyzation potentials were searched and their expression levels were profiled. Although RE was shown to happen in specific stages during *C. cinerea* development, the expression levels of putative RE enzymes were not consistent (Appendix A). Two tRNA specific adenosine deaminases (ADATs) were annotated in this fungus, and their expression levels remained steady in the life cycle, except that the expression level of one gene was six-fold higher in young fruiting bodies than in vegetative mycelia. We also profiled the expression levels of other deaminases. Guanine deaminase, AMP deaminase, and the cytosine deaminase-uracil phosphoribosyltransferase fusion protein showed higher expression levels in fruiting bodies than in mycelia. However, their expression levels in other stages were similar to or lower than that in mycelia. For the other three enzymes, cytidine deaminase, adenosine deaminase, and adenosine deaminase-like protein, vegetative mycelia had the highest expression level and the expressions in all other stages were downregulated. No putative adenosine deaminase acting on the RNA (ADAR) family ortholog and U-to-C aminases ortholog was identified.

The dominant type of RE reported in *G. lucidum* and *C. cinerea* here, namely U-to-C editing, was also observed in the nuclear and organelles of plants, including the model species of *Arabidopsis thaliana* [73,74]. In *Arabidopsis*, it was validated with transgenic plants that the edited gene sequence showed a higher transcript decay rate than the wild-type or non-edited genes, suggesting that the U-to-C editing affected the stability of mRNA due to the changes in the structure of mRNA [75]. In contrast to the case here, U-to-C editing is rare and C-to-U editing is common in plants. The RNA editosome pentatricopeptide repeat proteins (PPR) and DYW domains were thought to be involved in the C-to-U editing process in plant organelles, and that an unusual regulation principle led to U-to-C editing [76]. The synthetic PPR-DYW protein showed functional activities of U-to-C editing in bacteria and human culture cells [77]. The U-to-C RNA editing was validated in *C. cinerea* and *G. lucidum*; however, the PPR-DYW ortholog or functional domains were not identified, leaving the editing mechanism in these *Basidiomycota* largely unclear.

### 3.9. Alternative Splicing and RNA Editing Together Provided Adaptations in Development

The transcriptome profiles of different developmental processes and stages differed in terms of their evolutionary features, including the transcriptome age and divergence. In the sexual reproductive circle, from basidiospore germination, vegetative growth, to fruiting body formation, the trend of alternative splicing intensity aligned with the TAI and TDI profiles, but the trend of RNA editing intensity was opposite to the TAI and TDI profiles (Figure 6a). The highest value of AS and RE intensity was found in fresh basidiospores and sclerotia, two out of the three reproductive structures. Oidia possessed the least number of modifications among these three structures. After grouping the genes according to their ages, both AS transitions and RE were more commonly observed in old and conserved genes shared by eukaryotes. The non-synonymous CDS alterations decreased from old genes to young genes, while the modifications in non-coding and regulatory regions were more evenly distributed across gene ages (Figure 6b,c).

AS and RE, together, altered the transcriptomes in *C. cinerea* development and provided extra flexibility with a preference for evolutionarily old and conserved genes. Oidia, basidiospores, and sclerotia are three reproductive structures formed under different conditions, and they have different degrees of resistance to environmental stresses. The formation of fruiting bodies and the production of basidiospores require the genetic basis of hyphal mating between two mating types and meiosis [3]. Such sexual propagation process results in high genotypic plasticity, and basidiospores are fit for long distance travelling and carry greater genetic diversity for adapting to new habitats [78,79]. Transcripts in basidiospores are mainly produced during fruiting body maturation. RNA isoforms allow for the early responses to changing environments. Sclerotia need to survive under extreme and stressful environments for long periods of time [80]. They can be monokaryotic and dikaryotic from submerged mycelia; thus, they are not generated from sexual propagation, and such mycelia lack genetic diversity [3]. Both RNA editing and transcript isoforms were enriched in sclerotia. These two modifications could rapidly diversify the transcriptome without changing the genomic sequences [81,82]. As RNA editing precedes splicing and may lead to widespread effects on splicing and other molecular interactions [83], it seems that the RNA editing is preferred to efficiently alter the transcriptome. *C. cinerea* may rely on these mechanisms to efficiently increase its fitness under stressful environments [7,11]. On the contrary, a large number of oidia are rapidly asexually produced under a temperate environment, striving to inhabit favorable environments [84]. The population genetic background is of low diversity, and alternative modifications at the transcriptome level is inactive. These results suggested that the original gene regulatory system is fixed in the evolutionary history and sufficient for the adaptation and development under these two conditions [18]. During the complex multicellularity construction, from branching hyphae to mycelia and fruiting bodies, the transcriptomic divergence gradually increased. An increased number of young genes were functioning along with morphogenesis [11,54]. This regulation is well-programmed, with little post-transcriptional modifications. Fungi acquired divergent strategies with different developmental paths to adapt to their changing environments.

## 4. Conclusions

The development of fungi is responsive to and dynamic under changing environments. We demonstrated that alternative splicing and RNA editing events were active and developmentally regulated in *C. cinerea*, providing plasticity to the transcriptome. The expression of genes appeared to be controlled by modulated networks at multiple regulatory layers of stages of gene expression. As a whole, this fungus applies different strategies to increase its adaptability in the rapidly changing environment, involving the transitions in functional and metabolic programs, alterations in genetic and transcriptomic divergences, morphogenesis, and physiology.

## Figures and Tables

**Figure 1 jof-09-00915-f001:**
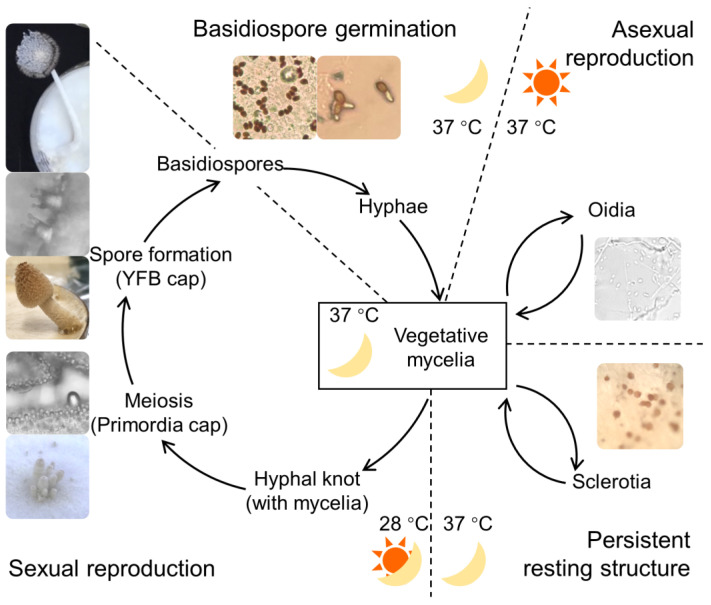
Life history of *Coprinopsis cinerea A43mutB43mut pab1-1* #326. Sun indicates continuous light incubation; moon indicates continuous dark incubation; sun/moon indicates the 12 h:12 h light–dark shift.

**Figure 2 jof-09-00915-f002:**
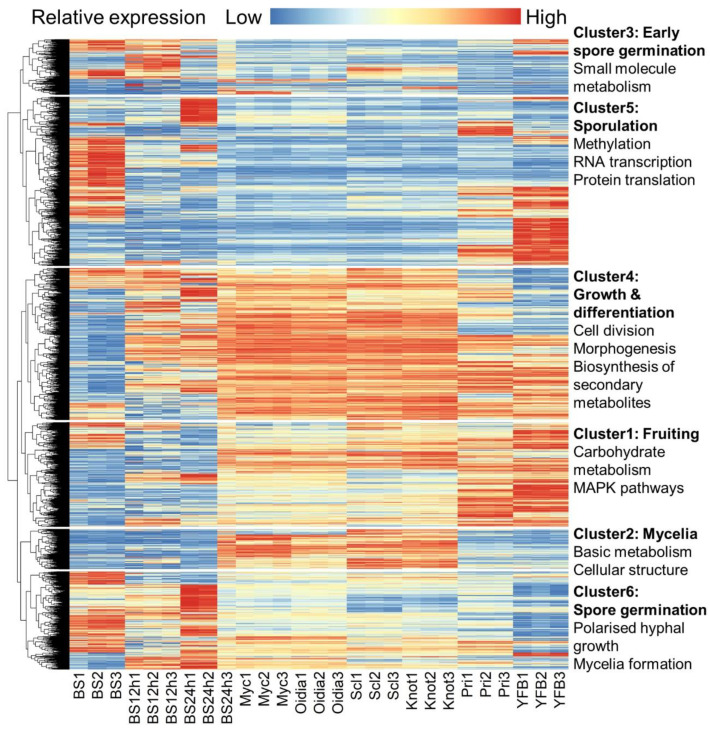
Heatmap of the gene expression and functional annotations of DEGs. TMM normalization were performed on the gene expression levels.

**Figure 3 jof-09-00915-f003:**
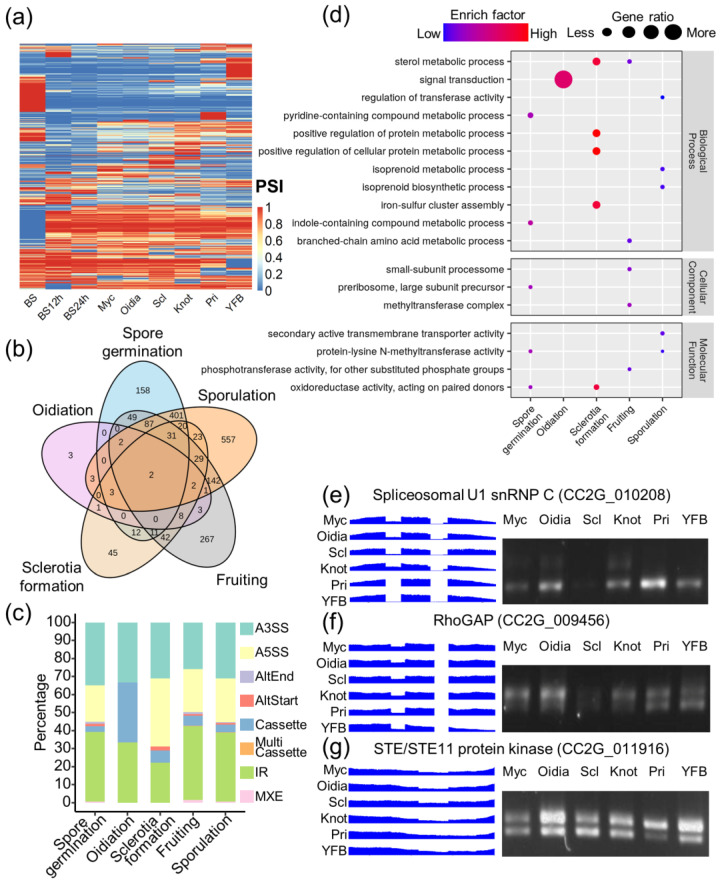
Landscape of alternative splicing in *C. cinerea* development. (**a**) Heatmap of PSI in differentially spliced sites; (**b**) summary of differentially spliced events in five developmental processes: (i) spore germination: BS-BS12h-BS24h; (ii) oidiation: Myc-Oidia; (iii) sclerotia formation: Myc-Scl; (iv) fruiting: Myc-Knot-Pri; and (v) sporulation: Pri-YFB-BS; (**c**) splicing types of differential spliced events; (**d**) GO annotation and enrichment of AS genes; and (**e**–**g**) RT-PCR validation of three AS events that were developmentally regulated.

**Figure 4 jof-09-00915-f004:**
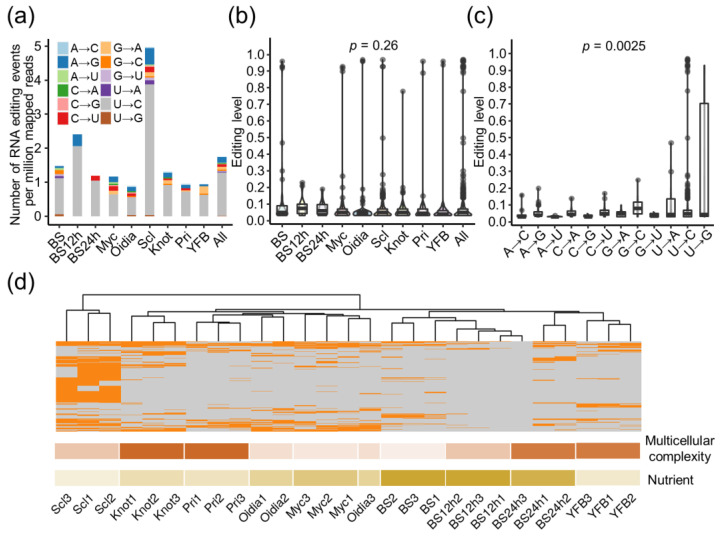
Landscape of RNA editing in *C. cinerea* development. (**a**) Number of editing events with different editing types found at nine developmental stages. (**b**) Distribution of editing level at different stages. (**c**) Distribution of editing level by editing types. Statistical results of (**b**,**c**) were detailed in Appendix A. (**d**) Presence or absence of RNA editing in each biological sample. Editing events were shaded in orange; the darker color below indicated much intensive morphogenetic changes or more nutrients.

**Figure 5 jof-09-00915-f005:**
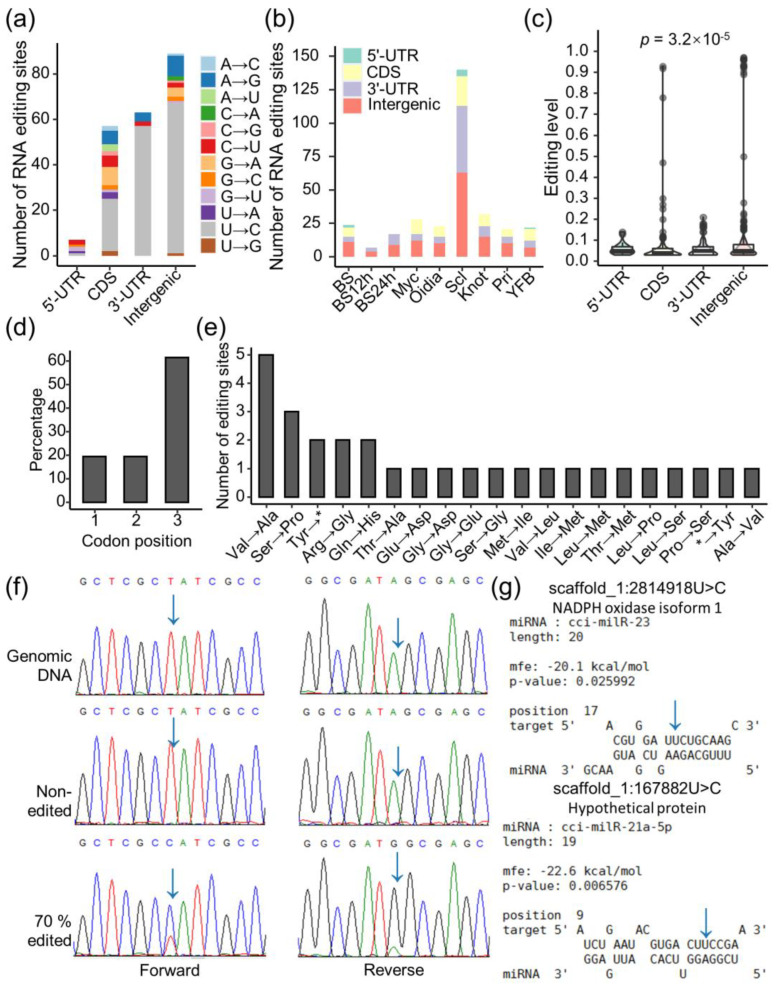
Functional impacts of RNA editing in *C. cinerea*. (**a**) Distribution of RNA editing sites in the genome. (**b**) Distribution editing sites found in nine developmental stages. (**c**) Editing levels of RNA editing sites at different genetic locations. (**d**) Frequency of codon change at three codon positions. (**e**) Missense RNA editing sites that result in amino acid changes or stop gain/loss. (**f**) Validation of predicted RNA editing site via Sanger sequencing. Arrows indicate the modified bases. Full images of all biological replicates were shown in Appendix A. (**g**) Examples of milRNA binding lost caused by RNA editing.

**Figure 6 jof-09-00915-f006:**
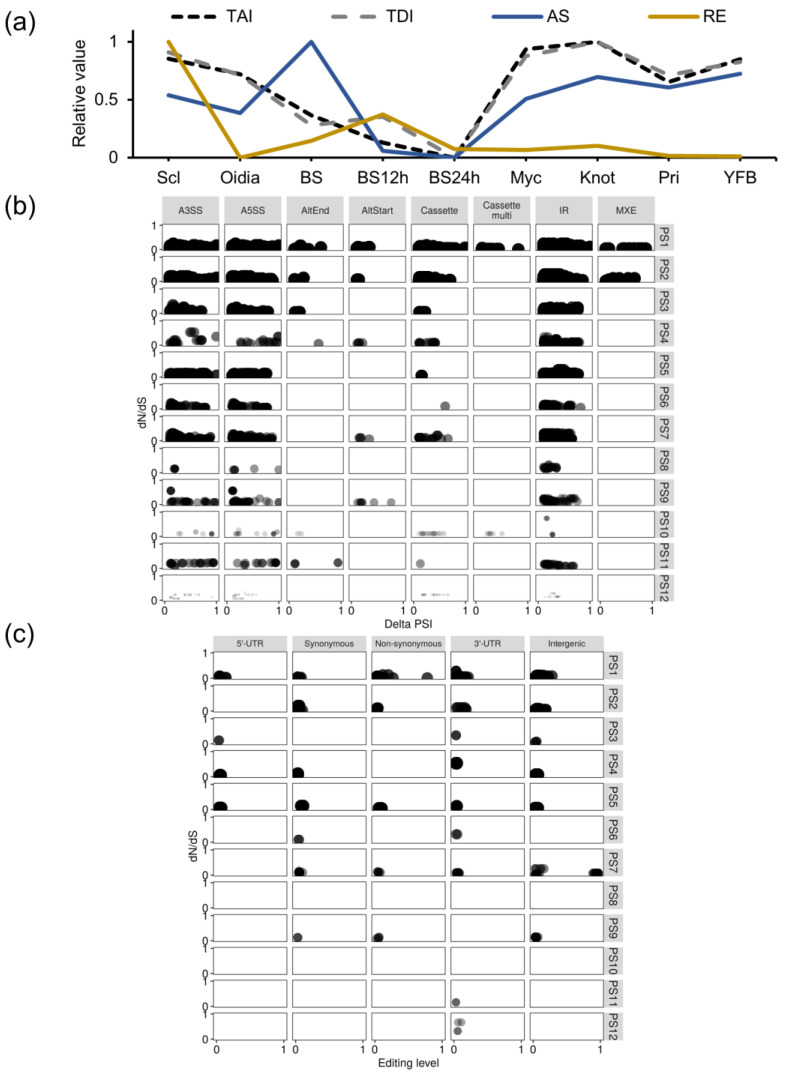
RNA modifications and phylotranscriptomic profiles in *C. cinerea* life history. (**a**) Transcriptome age index (TAI) and transcriptome divergence index (TDI), and numbers of alternative splicing (AS) and RNA editing (RE) in *C. cinerea* development. (**b**) Alternative splicing transitions are evaluated via delta PSI values; (**c**) RNA editing evaluated based on editing levels. Genes are assigned to phylostrata from PS1 (old) to PS12 (young). Darkness and circle size are scaled by the proportion of modified genes to the genome background for each phylostratum.

**Table 1 jof-09-00915-t001:** Strain cultivation and samples used for sequencing.

Stage (abb.)	Incubation
**Vegetative mycelia (Myc)**	Continuous darkness at 37 °C for 4 d
**Oidia-forming mycelia (Oidia)**	Continuous light at 37 °C for 4 d
**Sclerotia-forming mycelia (Scl)**	Continuous darkness at 37 °C for 12 d
**Mycelia with hyphal knots (Knot)**	Continuous darkness at 37 °C for 5.5 d, and 12 h:12 h light–dark cycle for 1 d
**Primordia undergoing meiosis (Pri)**	Continuous darkness at 37 °C for 5.5 d, and 12 h:12 h light–dark cycle for 6 d
**Young fruiting bodies undergoing spore formation (YFBs)**	Continuous darkness at 37 °C for 5.5 d, and 12 h:12 h light–dark cycle for 6.5 d
**Mature basidiospores (BS)**	Basidiospore discharged from mature cap
**Half-germinating basidiospores (BS12h)**	Continuous darkness at 37 °C for 12 h, broth 150 rpm
**Fully germinated basidiospores (BS24h)**	Continuous darkness at 37 °C for 24 h, broth 150 rpm
**Genomic DNA**	Continuous darkness at 37 °C

**Table 2 jof-09-00915-t002:** Statistics on the alternative splicing and RNA editing sites identified during the development of *C. cinerea*.

Stage	Alternative Splicing Sites	RNA Editing Sites
**BS**	3239	24
**BS12h**	1972	7
**BS24h**	1892	17
**Myc**	2575	29
**Oidia**	2412	24
**Scl**	2619	141
**Knot**	2829	33
**Pri**	2710	21
**YFB**	2865	22
**Total**	**6839**	**217**

## Data Availability

Sequencing data from this study have been submitted to the NCBI Sequence Read Archive (SRA, http://www.ncbi.nlm.nih.gov/sra, accessed on 9 August 2023) under BioProject accession numbers PRJNA560226, PRJNA573619, PRJNA573620. Genome resources of strain #326 can be obtained from GenBank (https://www.ncbi.nlm.nih.gov/assembly/GCA_016772295.1, accessed on 9 August 2023) or https://github.com/xieyichun50/Coprinopsis_cinerea_326_genome, accessed on 9 August 2023. Data summarization, statistics, and visualization were carried out under the R environment (R Core Team, 2022) using customized scripts. Codes on bioinformatic analyses are distributed at https://github.com/xieyichun50/Alternative-splicing-RNA-editing-analysis, accessed on 9 August 2023.

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
