# Peer review of "The Genome-Wide Characterization of Alternative Splicing and RNA Editing in the Development of Coprinopsis cinerea"

_jof, 2023, doi:10.3390/jof9090915_

Round 1
Reviewer 1 Report
The study “The Genome-wide Characterisation of Alternative Splicing and RNA Editing in the Development of Coprinopsis cinerea” aimed to evaluate the effects of RNA modifications during fungus developmental stages using genome and transcriptomic data.
The subject is very interesting, and the authors presented a wealth of results and analyses, including a huge list of supplementary material. However, this is a highly complex subject, which involves many layers of knowledge, such as the environmental variables affecting the nine stages of fungus development (as time of cultivation and light/dark). Added to this, the complexity of factors (including the environmental mentioned) and mechanisms acting in the regulation of cellular machinery, such as those influencing RNA (AS and RE were the mainly sudied) during the life stages of C. cinerea.
My general opinion is that the manuscript is not mature yet, and needs major revision in writing to be published. Briefly, in the introduction the authors do not mention the methodology used in the study, it needs to be included. The results must be concise, I suggest that the authors select the most important data to be presented in the paper, in order to facilitate the reading and understanding. I am sure this will help in the discussion, which must be precise and clear. I also suggest that the volume of supplemental material be reduced. The discussion is the most deficient topic in my opinion, it seems to me that the lack of objectivity in presenting the results made the discussion difficult. Most of the discussion is vague, not connected to a specific result, and thus, the contributions of the study are not clear or even mentioned. It might be better to write results and discussion in one topic, so that the connection between the topics would get more evident.
In the attached manuscript there are some comments, and I hope they can guide the authors in the revision process.

The presentation and discussion of the data are the biggest problem in the article. The quality of the English seems adequate, but it can certainly be re-evaluated after the restructuring of the text, as recommended.
Author Response
The study “The Genome-wide Characterisation of Alternative Splicing and RNA Editing in the Development of Coprinopsis cinerea” aimed to evaluate the effects of RNA modifications during fungus developmental stages using genome and transcriptomic data.
The subject is very interesting, and the authors presented a wealth of results and analyses, including a huge list of supplementary material. However, this is a highly complex subject, which involves many layers of knowledge, such as the environmental variables affecting the nine stages of fungus development (as time of cultivation and light/dark). Added to this, the complexity of factors (including the environmental mentioned) and mechanisms acting in the regulation of cellular machinery, such as those influencing RNA (AS and RE were the mainly sudied) during the life stages of C. cinerea.
My general opinion is that the manuscript is not mature yet, and needs major revision in writing to be published. Briefly, in the introduction the authors do not mention the methodology used in the study, it needs to be included. The results must be concise, I suggest that the authors select the most important data to be presented in the paper, in order to facilitate the reading and understanding. I am sure this will help in the discussion, which must be precise and clear. I also suggest that the volume of supplemental material be reduced. The discussion is the most deficient topic in my opinion, it seems to me that the lack of objectivity in presenting the results made the discussion difficult. Most of the discussion is vague, not connected to a specific result, and thus, the contributions of the study are not clear or even mentioned. It might be better to write results and discussion in one topic, so that the connection between the topics would get more evident.
In the attached manuscript there are some comments, and I hope they can guide the authors in the revision process.
Response: We thank the reviewer on the valuable suggestion. In this study, we characterized the alternative splicing and RNA editing in several developmental processes of C. cinerea. We aimed to provide a global view on the regulation of gene expression in the development of C. cinerea. We have revised the MS to further clarify the gene expression profile, as well as AS and RE, and the relationships among these mechanisms in C. cinerea.
Abstract
- Line 12: methodology is lacking
Response: We are sorry that the methodology was not clear in the abstract. We added the information to briefly summarize the methodology.
“In this study, we investigated the transcriptional changes and modifications in C. cinerea during the processes of spore germination, vegetative growth, oidiation, sclerotia formation and fruiting body formation by inducing different developmental paths of the organism and profiling the transcriptomes using the high-throughput sequencing method.”
Introduction
- Line 54: this picture must be better explored. I think its important to describe it in more details, and take additional informations from it
Response: We expanded the description to further clarify the life history of C. cinerea.
“Under different environmental conditions, the fungus chooses one of the developmental strategies from vegetative growth, asexual reproduction, sexual reproduction, or stress resistant structure formation (Figure 1). Vegetative mycelia expand on the nutrient-rich substrate at 37 °C with continuous darkness; asexual reproduction of oidiation is induced under the nutrient-rich environment at 37 °C with illumination; sexual reproduction happens when the fungus is under nutrient depletion and exposing to low temperature with light-dark cycles; and the stress resistant structure formation of sclerotia develop under nutrient depletion and continuous darkness [3,7].”
Results
- Line 225: “cultural” change to “culture”
Response: We corrected it as “The amount of nutrients in the culture media”
- Line 230: delete “process”
Response: We corrected it as “The fungus culture then entered the fruiting process or sclerotia formation under specific combinations of temperature and lightening.”
- Line 285: “to compare” or “and compared”
Response: We rephased the sentence as “TMM normalisation are performed on gene expression levels.”
- Line 301: in what picture can the reader see that?
Response: We added the citation of Figure S7b and Table S5 here.
- Line 302: this last sentence does not fit here
Response: We rephased the last two sentences of the paragraph as
“During sporulation, C. cinerea got the mean PSI that was higher than other stages by 0.1, and the effect of AS was also observed in fresh spores (Figure S7b and Table S5).”
- Line 307: I cant see that very clear in the picture. Could you explain better or improve the image?
Response: We have improved the presentation. Figure 3a is a subset of Figure S7d which showing the PSI of differentially spliced sites only.
- Line 320-322: the gene expression profile were close to one another in mycelia? I do not see that in figure 2, could you clarify that?
Response: We are sorry that our previous description was not clear. We added the citation of Figure S2, Figure S3, Figure S11 and Figure S12 here. The feature cannot be significantly told from Figure 2 because it only listed out the differentially expressed genes. In the genome-wide expression profile, mycelia-type stages were close to one another.
- Line 323: I would mention Figure S12 here again
Response: We thank the reviewer for the suggestion. We added the citation of Figure S12 here.
- Line 337-347: this paragraph seems more a discussion, and references are missing here
Response: We understand that in silico analysis results without experimental validation is not complete for formal functional studies, making this paragraph seems like a discussion. This paragraph presented the formal analysis results using the methods as described in section 2.6. The functional studies on each of the alternative splicing sites are out-of-scope to this MS. We added the citation about the reference genome annotation here.
- Line 347-350: Mention Fig 3e again here
Response: We improved the Figure 3 and cited the figures accordingly.
- Line 352: italics “C. cinerea”
Response: We corrected the typo.
- Line 352: (a) the spliced events are not showed in the picture
Response: We changed the figure legend as “(a) Heatmap of PSI in differentially spliced sites”.
- Line 386-387: Since this is a comparison with the literature it would fit better in discussion section. Besides, it is necessary a Figure? I dont think so...
Response: We revisited the dataset of the previous study. These formal analyses were not performed and the results were not presented before, not even a similar RNA editing characterization. We believed that it would be more appropriate to place the content in “Results” rather than in the “Discussion” to better structure the MS.
- Line 398: avoid starting a sentence with numerals
Response: We thank the reviewer for the suggestion. We rephased the sentence.
- Line 418: delete “were”
Response: We corrected the gramma of the sentence.
- Line 419: avoid starting a sentence with numerals
Response: We rephased the sentence.
- Line 420: change “create” to “creating”, “lead” to “leading”
Response: We corrected the typo.
- Line 423: “the” 137 amino acids
Response: We corrected the gramma of the sentence.
- Line 424: delete “repeat”
Response: This is a gene name identified from the gene annotation.
- Line 429: “but not in the”
Response: We corrected the gramma of the sentence.
- Line 439: are you sure is S16?
Response: We are sorry that the figure was not correctly cited. It should be S19.
- Line 476: italics “C. cinerea”
Response: We corrected the typo.
Discussion
- Line 490-491: what matrices exactly?
Response: We rephased the statement as “The comparison matrices across different developmental paths and stages”.
- Line 492: vegetative growth, asexual, sexual reproduction
Response: We corrected the gramma of the sentence.
- Line 493-495: this sentence is not well connected with your results. Must be improved!
Response: We rephased the sentence. This sentence told that previous study summarized the common and conserved features and this study pointed out the divergent features as presented in the next sentence.
- Line 495-498: very confusing and vague that sentence. What specific results led to that conclusion?
Response: Results section 3.2, Table S4 and Fig 2, Figure S4-6 discussed the developmental path- and stage-specific expression features of C. cinerea.
- Line 502-510: you mention these studies but do not mention what are they about. Do they corroborate your findings?
Response: We rephased the paragraph to make it logical.
- Line 519-521: please mention how did get to this conclusion. if this is your result, why are you mentioning the reference 71?
Response: We rephased the paragraph to clarify our statement.
- Line 523-525: are you still talking about references 22 and 72?
Response: We rephased the sentence to clarify our statement. This sentence was showing our findings.
- Line 525-527: ok, and how is this connected with you findings?
Response: We found the high abundance of AS in YFB and fresh spores, and the Aspergillus study provided evidence on the function of AS in spores, which together inspiring our discussion of “function of AS to adaptation and survival”.
- Line 573, 576-578: aren't these sentences highlighted contradicting each other?
Response: “but it did not show any high activity during sexual reproduction” referred to the sexual reproduction and fruiting process, “both sexual spores (basidiospores) and asexual spores (oidia), which are expected to respond to a broad range of environmental conditions, showed a higher editing level than other stages” referred to the spores.
- Line 583: in this topic, none of the results obtained in the study were related to the literature mentioned.
Response: This section responded to section Results 3.7, where we analysed the evolutionary features of genes with AS and RE. Together with the literature we mentioned, we further discussed the conserved and flexible strategies on the environmental response of the organism.
- Line 613-614: formatting
Response: We thank the reviewer for pointing this out. We corrected the format.
Reviewer 2 Report
The reviewed work, titled: “The Genome-wide Characterisation of Alternative Splicing and RNA Editing in the Development of Coprinopsis cinerea” by Yichun XIE, Po-Lam Chan , Hoi-Shan Kwan , Jinhui Chang is an outstanding study that is focused on the transcriptional regulation of the growth and development of Coprinopsis cinerea. The authors particular focus was on alternative splicing and RNA editing throughout multiple cell types and growth stages, using a genetic background that facilitated study of each. This study was well conceived, well designed, and experiments were thoughtfully performed. The analysis and validation of their findings is appropriate and thorough. On the whole, the manuscript is very well written and organized – and this reviewer found it to be a pleasure to read. I have no major concerns about design, results (and their interpretation), or the conclusions that the authors have drawn. I believe that this would be of interest to a variety of researchers who are interested in fungal biology and development. I personally found the authors findings (and conclusions) regarding the transcriptional plasticity that RNA editing and alternative splicing allow to be fascinating!
Major comments/concerns.
I have no major concerns about the design, execution, or interpretation of this study.
Minor comments.
Introduction:
Double check proper formatting for scientific names. The following is a helpful resource if there is any question and ambiguity:
· Thines, M., Aoki, T., Crous, P.W. et al. Setting scientific names at all taxonomic ranks in italics facilitates their quick recognition in scientific papers. IMA Fungus 11, 25 (2020). https://doi.org/10.1186/s43008-020-00048-6
Line 40: Basidiomycota is a proper name and should be italicized.
Line 42: Agaricales is the name of an order and should be italicized.
Line 96: ascomycetes is a proper name, should be capitalized and italicized.
As a note, please double check the entire paper for this – it would make it easier to follow if the authors are referring to something specific or generic and colloquial. There were multiple instances where proper formatting would clarify the section.
Materials and Methods:
Table 1 is very clear and informative – however it should be edited to cut down on the wasted space by changing the formatting a spacing accordingly.
Line 139: please clarify if there were any changes to the manufacturer’s protocol followed when using the RNeasy kit during the RNA extraction process. Please provide the quantity of sample that was ultimately processed and sequenced in the details.
Results:
Line 287 – I do not personally believe that you need to include the formula for relative expression in the figure legend. Id move that to the Materials and Methods to streamline the reading of this figure.
Line 352 – C. cinerea is a proper name and should be italicized.
Line 473 – Figure 6 b and 6 c have axis labels that are too small to read, please increase the font size of them all.
Discussion:
Line 484: Agaricales is the name of an order and should be italicized.
Line 539-540: Ascomycota 539 and Basidiomycota, ascomycetes – proper names that should be formatted accordingly.
Very well written throughout. Minor issues with the formatting of scientific names.
Author Response
Major comments/concerns.
I have no major concerns about the design, execution, or interpretation of this study.
Minor comments.
Introduction:
- Double check proper formatting for scientific names. The following is a helpful resource if there is any question and ambiguity:
- Thines, M., Aoki, T., Crous, P.W. et al. Setting scientific names at all taxonomic ranks in italics facilitates their quick recognition in scientific papers. IMA Fungus 11, 25 (2020). https://doi.org/10.1186/s43008-020-00048-6
Line 40: Basidiomycota is a proper name and should be italicized.
Line 42: Agaricales is the name of an order and should be italicized.
Line 96: ascomycetes is a proper name, should be capitalized and italicized.
As a note, please double check the entire paper for this – it would make it easier to follow if the authors are referring to something specific or generic and colloquial. There were multiple instances where proper formatting would clarify the section.
Response: We thank the reviewer to point out the issue. We have corrected the typo and format accordingly.
Materials and Methods:
- Table 1 is very clear and informative – however it should be edited to cut down on the wasted space by changing the formatting a spacing accordingly.
Response: We have formatted the table and figures to have a better reading experience for the audience.
- Line 139: please clarify if there were any changes to the manufacturer’s protocol followed when using the RNeasy kit during the RNA extraction process. Please provide the quantity of sample that was ultimately processed and sequenced in the details.
Response: We are sorry that the information was not included in the initial submission. We modified the section 2.1 as follows:
“Total RNA from the cap of primordia and YFB was isolated using RNeasy Plant Mini Kit (Qiagen, Germany) following the manufacturer’s instruction after microscopic examination. Approximately 5 µg of total RNA of each sample was sent to the Beijing Genomics Institute (BGI, Shenzhen, China) for library construction and sequencing. RNA libraries were prepared using TruSeq RNA Sample Prep Kit v2 (Illumina, USA) and sequenced with Illumina HiSeq® 4000 at the 2 × 150 bp paired-end read mode, the same as in our previous studies [7,22].”
Results:
- Line 287 – I do not personally believe that you need to include the formula for relative expression in the figure legend. Id move that to the Materials and Methods to streamline the reading of this figure.
Response: We thank the reviewer for the suggestions. The formula for relative expression is moved to section 2.2.
“Gene expression levels were calculated using Stringtie [44] with the reference gene annotation of C. cinerea #326 [11]. The count matrix was then fed to R package ‘edgeR’ [45] for expression analysis and presented by Trimmed Mean of M-values (TMM). Those genes with count-per-million (CPM) over one in at least two out of three replicates per stage were regarded as expressed genes. For each gene i, the relative expression level RE of each sample k was calculated as [formula] using log2 transformed TMM values. Differentially expressed genes (DEG) were determined with the threshold of |log2 (Fold change) | > 2, which was a fourfold change, and adjusted P value threshold of lower than 0.05. ”
- Line 352 – C. cinerea is a proper name and should be italicized.
Response: We thank the reviewer to point out the issue. We have corrected the typo and format accordingly.
- Line 473 – Figure 6 b and 6 c have axis labels that are too small to read, please increase the font size of them all.
Response: We have formatted the table and figures to have a better reading experience for the audience.
Discussion:
- Line 484: Agaricales is the name of an order and should be italicized.
Response: We have corrected the typo and format accordingly.
- Line 539-540: Ascomycota 539 and Basidiomycota, ascomycetes – proper names that should be formatted accordingly.
Response: We have corrected the typo and format accordingly.

Reviewer 3 Report
The group by Xie et al. have used the Coprinopsis cinerea strain A43mutB43mut pab1-1 #326 to investigate the effect of alternative splicing and RNA editing on the transcriptome during nine different developmental stages of the fungal life cycle. The background and some genomic material originate from previous C. cinerea transcriptome studies (Xie et al. 2020) and in addition Xie et al. 2021 has been involved in resequencing A43mutB43mut pab1-1 #326. The work is extensive analysis of the occurrence of alternative splicing and RNA editing at different growth phases.
Heatmap of gene expression and functional annotations of DEGs clustered the genes in six groups each of them showing specific gene expression for the growth phase. The alternative splicing of genes was detected in all stages of life cycle analyzed, altogether in 3411 genes. The highest occurrence was in germinating basidiospores (12 h). The common occurrence of alternative splicing in different phases of life cycle Of C. cinerea confirm the results obtained with the filamentous basidiomycete Schizophyllum commune (Sci. Rep. 2016, 6). The present work extends it to concern spore germination, oidia and sclerotia and the basidiospore development.
The research on RNA editing has neither got as much attention in basidiomycetes as in ascomycetes so the results of the present work are new. The research identified 217 RE sites and the most commonly in sclerotia. “A total of 89 editing sites (41.0 397 %) were found in intergenic regions. 128 other editing sites were located at the coding regions (CDS) or untranslated regions (UTR) of 114 genes, and none was found in the intron”. In future it is interesting to see how much the RE affects gene expression, could a RE editing in the 5´upstream region affect the promoter activity of the gene.
The manuscript contains important new information that has to be considered in future when the expression of genes is investigated in filamentous basidiomycetes. The results are well- illustrated and the English text is clear. The manuscript is recommended for publication in the special issue Molecular Genetics and Genomics of Mushroom-Forming Fungi.
Small remarks
Table 1. present the nine stages investigated, at the end “genomic DNA” what does it mean?
I wonder about the regulation of alternative splicing, it seems that it is related to environmental factors , but how?
The same question about RNA editing does it happen by chance or is it regulated?
Is the regulation of fruiting of Coprinopsis cinerea haploid strain A43mutB43mut pab1-1 #326 similar to the fruiting dikaryotic strain created by mating? See Muraguchi ,et al. 2015| DOI:10.1371/journal.pone.0141586. The strain has been used for ages in lab, could that influence the recoded results? Of course, this also true for other filamentous basidiomycete strains cultured for a long time in lab.
Although the results are well and abundantly illustrated, a simple table with different life cycle stages and the abundance of alternative splicing and RNA editing at different phase could be easily seen would make the reader happy.
Author Response
- Table 1. present the nine stages investigated, at the end “genomic DNA” what does it mean?
Response: We are sorry that it was not clarified in the initial submission. Genomic DNA sequence was required to have a precise identification of RNA editing events, so such sample was collected and sequenced. We modified the presentation in section 2.1 as follows:
“Illumina short-read DNA sequences were acquired from our previous work of de novo genome assembly on strain #326 [11] to serve as the supporting genomic sequence information for the determination of RNA editing. In this study, all DNA and RNA samples were collected from the same batch of culture.”
- I wonder about the regulation of alternative splicing, it seems that it is related to environmental factors, but how?
Response: We thank the reviewer for asking this question. It is known that AS functions in the response against environmental factors, especially the stresses. Stress induces AS through splicing factors, and splicing factors are regulated at multiple levels, including the regulation of expression levels, type of isoforms, and the phosphorylation of predominantly serine/arginine-rich proteins and other splicing factors. To the upstream level, the regulation on splicing factors may be controlled by the epigenetic changes and central responsive kinases (see mini review - Rapid Regulation of Alternative Splicing in Response to Environmental Stresses, Liu et al., 2022, https://doi.org/10.3389/fpls.2022.832177). In this study, we find that the expression levels of splicing factors were different across stages, with gene usage preferences in different stages. However, due to the limiting information here, we do not include these discussions in the current manuscript.
- The same question about RNA editing does it happen by chance or is it regulated?
Response: We thank the reviewer for the question. From the cases found in Cephalopoda and Drosophila, RNA editing occurred both conserved and by chance, providing extra flexibility to the organism in environmental interactions. In the Ascomycota, RNA editing was developmentally regulated in the sexual reproduction of Fusarium and Neurospora. Due to the limiting information in Basidiomycota, it is hard to draw a conclusion now. We wish to perform a more comprehensive study of RNA editing in the future, which can recruit more species with several developmental stages and triggers of environmental factors, so that the conservation and diversification features can be compared and discussed.
- Is the regulation of fruiting of Coprinopsis cinerea haploid strain A43mutB43mut pab1-1 #326 similar to the fruiting dikaryotic strain created by mating? See Muraguchi ,et al. 2015| DOI:10.1371/journal.pone.0141586. The strain has been used for ages in lab, could that influence the recoded results? Of course, this also true for other filamentous basidiomycete strains cultured for a long time in lab.
Response: In the previous study presented by Muraguchi et al. (2015), fruiting was induced on a YM1/2G agar plate medium, with the incubation of five days in dark, followed by 2 hr in light, and 24 hr in dark. Tissues during the fruiting body development were then sampled with defined time interval. To the best of our knowledge, the hyphal knots stage, primordia stage and YFB stages were very close to the stage 2, stage 6-8 and stage 8-10 as described in Muraguchi et al., respectively, but not exactly the same. The regulations we observed here were similar to Muraguchi’s study. Due to the limited information, we are hard to tell whether the long-cultured time and history in lab will influence the recorded results or the adaptive response of the fungus.
- Although the results are well and abundantly illustrated, a simple table with different life cycle stages and the abundance of alternative splicing and RNA editing at different phase could be easily seen would make the reader happy.
Response: We thank the reviewer for the suggestion. We added “Table 2” to show the statistic on Alternative splicing and RNA editing sites identified during the development of C. cinerea.
Reviewer 4 Report
The authors characterize alternative splicing and RNA editing in Coprinopsis cinerea. After minor revisions, the paper would be acceptable for publication in JoF.
Minor points
In some sentences, the authors use “shall”.
When I asked this question to AI, AI answered as follows:
In scientific papers, the use of "shall" is generally not preferred. Instead, more precise and direct language is usually preferred, such as using "will" or "should" when discussing procedures or recommendations. However, it's important to follow the specific style guide or formatting requirements of the journal or publication you are submitting to, as they may have their own guidelines on the use of certain words.
Line 339: “of ” in front of “would” should be removed.
Line 352: “C. cinerea” should be italicized.
Line 613: The last sentence of this paragraph should be removed.
Author Response
Minor points
In some sentences, the authors use “shall”.
When I asked this question to AI, AI answered as follows:
In scientific papers, the use of "shall" is generally not preferred. Instead, more precise and direct language is usually preferred, such as using "will" or "should" when discussing procedures or recommendations. However, it's important to follow the specific style guide or formatting requirements of the journal or publication you are submitting to, as they may have their own guidelines on the use of certain words.
Response: We have revised the manuscript to use proper words.
Line 339: “of ” in front of “would” should be removed.
Response: We thank the reviewer to point out the issue. We have corrected the sentence accordingly.
Line 352: “C. cinerea” should be italicized.
Response: We thank the reviewer to point out the issue. We have corrected the typo and format accordingly.
Line 613: The last sentence of this paragraph should be removed.
Response: We thank the reviewer to point out the issue. We have corrected the typo and format accordingly.
Round 2
Reviewer 1 Report
The authors made the punctual changes suggested in the text, and that's ok. However, I mentioned that, in addition to the markings, it was necessary to improve the manuscript, that is, to restructure the text to make it more direct and simple. And this was not done. It was also suggested to reduce the supplementary material, and in the response to the question below (just to illustrate), what happened was a greater citation of the supplementary material.
- Line 320-322: the gene expression profile were close to one another in mycelia? I do not see that in figure 2, could you clarify that?
Response: We are sorry that our previous description was not clear. We added the citation of Figure S2, Figure S3, Figure S11 and Figure S12 here. The feature cannot be significantly told from Figure 2 because it only listed out the differentially expressed genes. In the genome-wide expression profile, mycelia-type stages were close to one another.
Still, I would like to ask the authors attention to the number of the quoted lines in the answers. There was a change in relation to the first version of the manuscript, and it does not correspond to what is marked in the text. I ask the authors to review the lines number, and after careful observation of all points, then resubmit the manuscript.
I reinforce my previous comment. The subject is complex, and therefore needs to be exposed in a more direct and simple way. The text, as it stands, is confusing and exhausting. The restructuring of the results and discussion as suggested in my previous review (see below) was not done in the revised version.
My general opinion is that the manuscript is not mature yet, and needs major revision in writing to be published. The results must be concise, I suggest that the authors select the most important data to be presented in the paper, in order to facilitate the reading and understanding. I am sure this will help in the discussion, which must be precise and clear. I also suggest that the volume of supplementary material be reduced. The discussion is the most deficient topic in my opinion, it seems to me that the lack of objectivity in presenting the results made the discussion difficult. Most of the discussion is vague, not connected to a specific result, and thus, the contributions of the study are not clear or even mentioned. It might be better to write results and discussion in one topic, so that the connection between the topics would get more evident.
Minor editing of English language required
Author Response
We Thank the reviewer for the valuable comments. In this submission, we restructured the manuscript by combining the results and discussion, reconsidered the points to be discussed, and we cut down the supplementary materials. Since there were great changes, we only uploaded a clean version of the MS here.
Round 3
Reviewer 1 Report
The quality of the paper has improved considerably in this version. Congratulations to the authors!